

# Evaluation of the diagnostic and prognostic values of serum HSP90α in sepsis patients: a retrospective study

Fuxing Li[1,*], Yulin Zhang[1,*], Bocheng Yu[1], Zihua Zhang[1], Yujuan Fan[1], Li Wang[1], Mingjing Cheng[1], Ping Yan[2] and Weidong Zhao[1,3]

[1] Department of Clinical Laboratory, School of Clinical Medicine, Dali University, Dali, Yunnan, China
[2] Department of Gastroenterology, The First Affiliated Hospital of Dali University, Dali, Yunnan, China
[3] Institute of Translational Medicine for Metabolic Diseases, Dali University, Dali, Yunnan, China
* These authors contributed equally to this work.

Corresponding authors
Ping Yan, yanping@dali.edu.cn
Weidong Zhao, wdzhao@dali.edu.cn

## ABSTRACT

**Background:** Sepsis is a serious syndrome that is caused by immune responses dysfunction and leads to high mortality. The abilities of heat shock protein 90α (HSP90α) in assessing the diagnosis and prognosis in patients with sepsis remain ill-defined to date. We conducted a study to reveal the possible clinical applications of HSP90α as biomarker for the diagnosis and prognosis in patients with sepsis.
**Methods:** In total, 150 patients of sepsis, 110 patients without sepsis admitted to ICU and 110 healthy subjects were involved in this study. The serum HSP90α contents, sequential organ failure assessment (SOFA) scores, procalcitonin (PCT), and short-term survival status of the participants were measured and compared. Logistic and linear regression models adjusting for potential confounders were used to examine the association of HSP90α with sepsis survival. Moreover, serum IL-1β, IL-18, MIP-3α, and ENA-78 were also determined. Finally, Spearman correlation analysis was employed to reveal a possible mechanism that HSP90α contributed to the short-term deaths.
**Results:** Serum HSP90α levels in sepsis patients were higher than those in ICU controls and healthy controls ($P < 0.001$), and even increased in patients who died within 28 days ($P < 0.001$). Logistic and linear regression models identified HSP90α was an independent risk factors for sepsis mortality. Receiver operating characteristic (ROC) analysis displayed that HSP90α had a considerable predictive performance for sepsis outcome, with an area under curve (AUC) value up to 0.79. Survival analysis demonstrated that the mortality of sepsis individuals at 28 days was positively associated with HSP90α levels, especially the levels of HSP90α were greater than 120 ng/mL ($P < 0.001$). Moreover, among sepsis patients, those who died had notably elevated cytokines, IL-1β, IL-18, and chemokines, MIP-3α, ENA-78, relative to survivors. Further correlation analysis demonstrated that there was a nominally positive correlation between HSP90α and IL-1β, IL-18, and MIP-3α.
**Conclusion:** HSP90α is of favorable clinical significance in sepsis diagnosis and prognosis, laying a foundation for future clinical applications.

## INTRODUCTION

Sepsis syndrome represents a major global health issue in today's medicine (*Coopersmith et al., 2018*). Annually, an estimated 20 to 50 million people have sepsis worldwide, with 10 million death (*Rudd et al., 2020*). Despite great advances in the therapeutic strategies of sepsis (*Fleuren et al., 2020*; *Papafilippou et al., 2021*; *Uffen et al., 2021*), the mortality of patients with sepsis remains to be discouraging (*Grebenchikov & Kuzovlev, 2021*).

Since the high mortality of sepsis attribute to the untypical clinical manifestation, early diagnosis of sepsis may be of clinical significance and can save numerous lives (*Daly et al., 2020*). In this context, some laboratory biomarkers, such as procalcitonin (PCT) (*Vijayan et al., 2017*), C reactive protein (CRP) (*Hofer et al., 2012*), interleukin-6 (IL-6) (*Ma et al., 2016*), serum amyloid A (SAA) (*Arnon et al., 2007*), and heparin-binding protein (HBP) (*Yang et al., 2019*), were widely applied in the diagnosis and predicting outcomes of sepsis in the clinical practice, but with the unsatisfied sensitivity and specificity. Thus, recent reports have suggested the use of some new biomarkers for the diagnosing and predicting clinical outcomes of sepsis (*Pierrakos et al., 2020*).

Heat shock proteins (HSPs) are highly conserved polypeptides and essential for diverse cellular processes, such as recognition, signaling transduction, protein maturation, and cell differentiation (*Morimoto, 1998*). In particular, many recent studies have indicated that HSP90α is strongly related to the proliferation and differentiation in cancer cells (*Secli et al., 2021*; *Zavareh et al., 2021*; *Zhang et al., 2021*). To our knowledge, few reports regarding HSP90α in sepsis have been published. *Fitrolaki et al. (2016)* reported that elevated HSP90α contributed to acute inflammatory metabolic dysfunction and multiple organ failure in pediatric sepsis. In animal models, blockage of HSP90α could attenuate the inflammatory damage (*Kubra et al., 2020*; *Li et al., 2017*; *Zhao et al., 2013*). Nonetheless, these therapies have not advanced to the clinical applications. Moreover, studies on the diagnosis and prognosis of HSP90α in sepsis remain limited.

With these issues in mind, we evaluated whether the presence of serum HSP90α on ICU admission is a potential predictor for the early-onset of sepsis and a key indicator for survival stratification. Furthermore, we explored a possible underlying mechanism that HSP90α affect the clinical outcome of sepsis patients by eliciting the secretion of pro-inflammatory cytokines and chemokines.

## MATERIALS AND METHODS

### Study subjects

In total, 150 patients diagnosed with sepsis in the First Affiliated Hospital of Dali University (a tertiary grade A public hospital with 1,000 beds) between September 2019 and September 2020 were enrolled, including 101 males, 49 females, with an average age of (57.95 ± 15.39) years. All subjects were enrolled in agreement with the diagnostic standard described in Sepsis 3.0. A total of 110 patients treated in the ICU without sepsis or

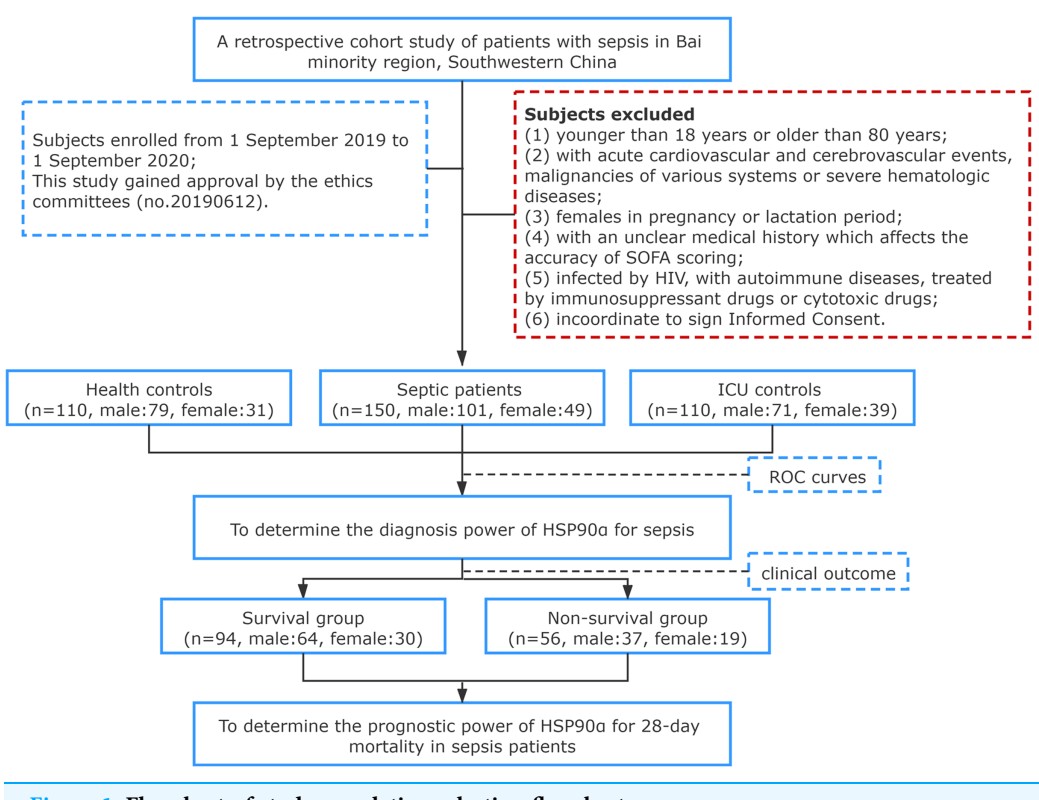

**Figure 1 Flowchart of study population selection flowchart.**

signs of infection served as controls, including 71 males, 39 females, with an average age of (56.12 ± 10.52) years. Healthy individuals for control were from the physical examination center of our hospital, including 79 males and 31 females, with an average age of (58.38 ± 15.10) years. According to clinical outcome, the 150 sepsis patients were divided into two groups: Survival group ($n$ = 94), including 64 males and 30 females, with an age of (57.62 ± 15.68) years; Death group, including 37 males and 19 females, with an age of (58.38 ± 15.10) years. Patients who met any of the following criteria were excluded: (1) younger than 18 years or older than 80 years; (2) with acute cardiovascular and cerebrovascular events, malignancies of various systems or severe hematologic diseases; (3) females in pregnancy or lactation period; (4) with an unclear medical history which affects the accuracy of SOFA scoring; (5) infected by HIV, with autoimmune diseases, treated by immunosuppressant drugs or cytotoxic drugs; (6) incoordinate to sign Informed Consent. All subjects we induced in this study were informed of the research content and signed informed consent, and the First Affiliated Hospital of Dali University granted Ethical approval to carry out the study within its facilities (Ethical Application Ref: 20190612) (no. 20190612) (Fig. 1).

## Data collection

The hospital information system (HIS) was consulted to refer to and collect clinical pathological data of patients based on case-control approach. Data recorded were as follows: (1) general information, such as age and gender; (2) clinical manifestations,

including infection route, complications, hospital stays, vital signs (body temperature, respiratory rate, heart rate, blood pressure); (2) laboratory indexes, including blood routine, total bilirubin (TBI, normal value: 5.1–19.0 μmol/L), creatinine (Crea, normal value: 22–123 μmol/L), blood urea nitrogen (BUN, normal value: 2.86–8.20 mmol/L), lactic acid (Lct, normal value: 0.5–2.2 mmol/L), C-reactive protein (CRP, normal value: <10 mg/L), procalcitonin (PCT, normal value: <0.5 ng/mL), and SOFA score, which were obtained from the examinations on peripheral venous blood samples within 48 h of admission. However, the data of CRP and PCT levels, SOFA cores, ICU stay, as well as 28-day mortality of health controls were not collected, because the levels of CRP and PCT of health individuals were not tested, and no health subject was admitted to the ICU.

## Serum HSP90α levels measurement

Venous blood samples (5 mL) were collected from all subjects with an empty stomach within 48 h of admission, and then preserved in a −80 °C refrigerator following serum separation. ELISA method was applied to determine serum HSP90α level. In short, sample buffer was firstly taken to dilute the standard, quality control and serum samples, after which the diluted samples were transferred to a 96-well micro-plate. Horseradish peroxidase-conjugated antibodies were then added, and tetramethyl benzidine was used to observe the reaction after 1 h of incubation with a temperature of 37 °C. Finally, a microplate reader was used to read the optical density (OD) value of each well at 450 nm in wavelength, and corresponding standard curve was plotted to further calculate the content of HSP90α in each group.

## Cytokines and chemokines measurement

Cytokines and chemokines quantification were completed using multi-analyte flow assay LEGENDplex™ Human Cytokine Panel 2 kit (Cat. 740102; BioLegend, San Diego, CA, USA) and LEGENDplex™ Human Proinflammatory Chemokine Panel (Cat. 740003; BioLegend, San Diego, CA, USA). FACS analysis was carried out with a FACS Calibur flow cytometer (BD Biosciences, San Jose, CA, USA).

## Statistics

All statistical analyses were performed with SPSS25.0 for windows (IBM, NY, USA), figures were generated by GraphPad Prism 7.0 (GraphPad Prism, San Diego, CA, USA) and R package pheatmap (v1.0.12). All measurement data were expressed as mean ± standard deviation (SD). One-way analysis of variance for date of normal distribution, for data of non-normal distribution, Mann–Whitney U test was implemented for between-group comparison. For numeration data, $\chi^2$ or Fisher's exact test was adopted for between group comparison. To determine the diagnosis power of HSP90α, PCT and SOFA score for sepsis, receiver-operating characteristic (ROC) curves were constructed and the area under the curve (AUC) was calculated with its 95% confidence interval (CI). Spearman correlation analysis was conducted to elucidate the association of HSP90α with SOFA score, PCT. Then univariate logistic regression was employed to investigate the effect of various parameters on survival in patients with sepsis by adjusted

**Table 1 Demographic, clinical, and laboratory profiles of septic patients, ICU controls and healthy controls.**

| Parameter | Health controls ($n = 110$) | ICU controls ($n = 110$) | Sepsis patients ($n = 150$) |
|---|---|---|---|
| Patient characteristics | | | |
| Age, years | 57.34 ± 11.47 | 56.12 ± 10.52 | 57.95 ± 15.39 |
| Male sex | 79 (63.6%) | 71 (64.5%) | 101 (67.3%) |
| Laboratory values | | | |
| WBC (×10$^9$/L) | 6.15 ± 1.59 | 9.58 ± 4.20[a] | 13.46 ± 9.01[a,b] |
| N (×10$^9$/L) | 3.75 ± 1.28 | 7.62 ± 4.00[a] | 10.78 ± 5.99[a,b] |
| L (×10$^9$/L) | 1.92 ± 0.45 | 1.35 ± 0.82[a] | 0.94 ± 0.74[a,b] |
| M (×10$^9$/L) | 0.36 ± 0.13 | 0.94 ± 0.22[a] | 0.70 ± 0.52[a] |
| HCT | 43.48 ± 3.36 | 41.11 ± 8.58 | 35.36 ± 9.17[a,b] |
| PLT (×10$^9$/L) | 234.94 ± 49.62 | 198.99 ± 91.80[a] | 137.30 ± 100.30[a,b] |
| CRP, mg/L | NA | 97.33 ± 95.41 | 109.68 ± 78.68[b] |
| PCT, ng/ml | NA | 6.39 ± 16.19 | 15.27 ± 28.57[b] |
| HSP90α (ng/mL) | 19.38 ± 16.70 | 123.16 ± 94.14[a] | 242.07 ± 215.70[a,b] |
| SOFA score | NA | 1.52 ± 0.63 | 6.10 ± 3.86[b] |
| ICU stay, days | NA | 3.42 ± 1.56 | 15.86 ± 13.19[b] |
| 28-day mortality | NA | 0 (0%) | 56 (37.3%)[b] |

Notes:
Continuous values as mean ± standard deviation, categorical values as absolute number and percentage. WBC, white blood cell count; N, neutrophils; L, lymphocyte; M, monocytes; HCT, hematocrit value; PLT, platelets; CRP, C-reaction protein; PCT, procalcitonin; SOFA, sequential organ failure assessment; ICU, intensive care unit; NA, not applicable.
[a] Differences were considered statistically significant with $P < 0.05$ compared to the health controls.
[b] Differences were considered statistically significant with $P < 0.05$ compared to the ICU controls.

odds ratios (ORs). We also constructed five different models. Model 1 included only HSP90α. PCT was added to Model 2. Model 3 was Model 2 with further adjustment for SOFA score. Model 4 was Model 3 with further adjustment for creatinine. Finally, more potential indicators and confounders, including urea nitrogen was added to Model 4. The mortality in 28 days was defined as the endpoint, ROC analyses and area under the ROC curve (AUC) values were used to evaluate creatinine (Crea), blood urea nitrogen (BUN), SOFA score, PCT, and HSP90α prognostic power for 28-day mortality. Kaplan–Meier analysis was implemented for the survival of sepsis patients of various groups. Finally, Spearman correlation analysis was employed to evaluate the correlation between HSP90α with IL-1β, IL-18, ENA-78, and MIP-3α. $P < 0.05$ refers to between-group difference of statistical significance.

# RESULTS

## Subjects characteristics

The demographic and laboratory profiles of the studied subjects are shown in Table 1. In total, 150 patients with sepsis (57.95 ± 15.39 years) were enrolled in this study, with 101 men (67.3%). A total of 110 sex- (men, 63.6%) and age-matched (57.34 ± 11.47, years) health individuals were included as health controls, and 110 sex- (men, 64.5%) and age-matched (56.12 ± 10.52, years) ICU non-sepsis patients were selected as ICU controls. Higher numbers of white blood cells (WBC), neutrophils, monocytes, lower numbers of

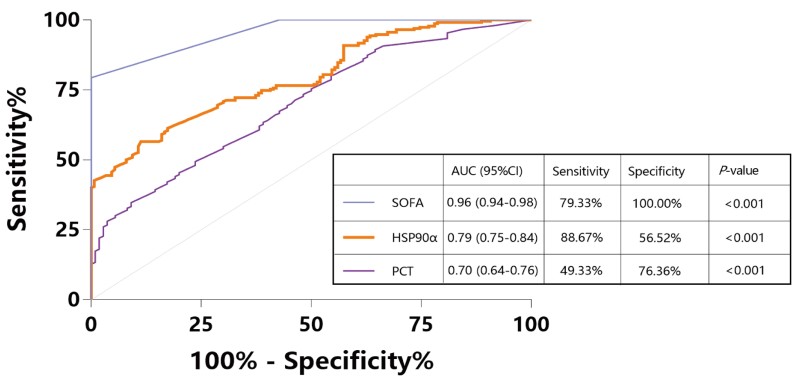

**Figure 2 Receiver operating characteristic curve (ROC) of HSP90α, PCT and SOFA score for diagnosis of sepsis.** HSP90α, heat shock protein 90α; PCT, procalcitonin; SOFA, Sequential Organ Failure Assessment.          

lymphocytes, platelets, and lower ratio of hematocrit were observed in the patients with sepsis (all $P < 0.001$). Compared with health controls and ICU controls, serum HSP90α concentrations at admission were significantly increased in the patients with sepsis ($P < 0.001$).

## Serum level of HSP90α as a potential diagnostic biomarker for the patients with sepsis

To assess the usefulness of HSP90α as a diagnostic marker for sepsis, we pooled 150 sepsis patients and 110 ICU non-sepsis patients as controls for HSP90α, SOFA and PCT, and the ROC analysis was applied. The AUC of HSP90α was 0.79 (95% CI [0.75–0.84], $P < 0.001$) with 88.67% sensitivity and 56.52% specificity in discriminating the patients with sepsis, which was weaker than SOFA score (AUC: 0.96, 95% CI [0.94–0.98], 79.33% sensitivity and 100% specificity) and superior than PCT (AUC: 0.70, 95% CI [0.64–0.76], 49.33% sensitivity and 76.36% specificity) (Fig. 2). These results reveal that HSP90α can be used as a potential diagnostic biomarker for the patients with sepsis.

## Elevated serum HSP90α level in non-survivals of sepsis patients

Next, to investigate its ability on the differential of outcomes of sepsis, the serum HSP90α levels were determined in both survivors and non-survivors of sepsis group. The characterization of the survivals and non-survivals of sepsis individuals is presented in Table 2. No statistical differences in age, sex, etiology, comorbidities, hospital stays, and constants were observed in both survivors and non-survivors (all $P > 0.05$). In addition, there were no significant differences in WBC, neutrophils, lymphocytes, C-reactive protein, total bilirubin, PaO$_2$, PaCO$_2$, and lactate (all $P > 0.05$). However, non-survivors had significantly higher levels of procalcitonin (PCT, $P = 0.003$), higher SOFA score ($P < 0.001$), higher Crea levels ($P = 0.006$), higher BUN levels (=0.009), and higher serum HSP90α levels ($P < 0.001$) than survivors. Moreover, the spearman correlation analysis of all patients with sepsis yielded a positive correlation of serum HSP90α levels at ICU admission with SOFA score ($r = 0.443$, $P < 0.001$, Fig. 3A) and PCT ($r = 0.373$, $P < 0.001$, Fig. 3B), respectively.

**Table 2 Index comparison between the survival and non-survival groups of sepsis cases.**

| Parameter | Survival ($n$ = 94) | Non-survival ($n$ = 56) | $P$-value |
|---|---|---|---|
| Patient characteristics | | | |
| Age, years | 57.62 ± 15.68 | 58.38 ± 15.10 | 0.774 |
| Male sex | 64 (68.1%) | 37 (66.1%) | 0.799 |
| Etiology | | | |
| Pulmonary infection | 20 (21.3%) | 17 (30.4%) | 0.212 |
| Acute pancreatitis | 7 (7.5%) | 3 (5.4%) | 0.875 |
| Postoperative infection | 10 (10.6%) | 6 (10.7%) | 0.988 |
| Others | 57 (60.6%) | 30 (53.6%) | 0.058 |
| Comorbidities | | | |
| Hypertension | 32 (34.0%) | 22 (39.3%) | 0.926 |
| Diabetes | 12 (12.8%) | 12 (21.4%) | 0.162 |
| COPD | 21 (22.3%) | 16 (28.6%) | 0.392 |
| Hospital stays, days | 16.12 ± 11.29 | 15.87 ± 16.35 | 0.077 |
| Constants | | | |
| Pulse rate, beats/min | 94.45 ± 19.62 | 101.75 ± 20.90 | 0.056 |
| Breath rate, beats/min | 20.56 ± 4.21 | 20.42 ± 4.45 | 0.935 |
| Systolic blood pressure, mmHg | 119.82 ± 19.56 | 121.20 ± 22.58 | 0.605 |
| Diastolic blood pressure, mmHg | 73.88 ± 14.36 | 71.78 ± 13.84 | 0.787 |
| Laboratory values | | | |
| WBC ($\times 10^9$/L) | 12.47 ± 6.59 | 16.35 ± 14.73 | 0.144 |
| N ($\times 10^9$/L) | 10.29 ± 5.84 | 11.20 ± 5.52 | 0.252 |
| L ($\times 10^9$/L) | 1.02 ± 0.87 | 0.98 ± 1.19 | 0.409 |
| CRP (mg/L) | 101.60 ± 71.48 | 128.38 ± 86.56 | 0.088 |
| PCT (ng/mL) | 11.08 ± 23.97 | 22.73 ± 34.23 | 0.003 |
| SOFA (score) | 5.01 ± 3.17 | 7.89 ± 4.27 | <0.001 |
| TBI (μmol/L) | 33.81 ± 38.84 | 42.77 ± 66.04 | 0.766 |
| Crea (μmol/L) | 145.34 ± 159.13 | 213.53 ± 204.17 | 0.006 |
| BUN (mmol/L) | 10.35 ± 8.36 | 14.32 ± 10.83 | 0.009 |
| $PaO_2$ (mmHg) | 95.58 ± 71.81 | 76.09 ± 34.33 | 0.321 |
| $PaCO_2$ (mmHg) | 34.69 ± 6.90 | 40.59 ± 18.37 | 0.348 |
| Lct (mmol/L) | 2.85 ± 2.96 | 3.68 ± 4.29 | 0.191 |
| HSP90α (ng/mL) | 135.88 ± 116.90 | 373.35 ± 263.29 | <0.001 |

Note:
Continuous values as mean ± standard deviation, categorical values as absolute number and percentage. WBC, white blood cell count; N, neutrophils; L, lymphocyte; CRP, C-reactive protein; PCT, procalcitonin.

## HSP90α is related to the sepsis deaths

In order to accurately identify the risk factors for survival in patients with sepsis, we performed the univariate logistic regression analysis. As shown in Fig. 4, five variables were identified as significant risk factors for the survival of sepsis patients, including PCT, SOFA, creatinine (Crea), blood urea nitrogen (BUN) and HSP90α.

Furthermore, five different multivariable models were constructed to explore the relationship between HSP90α and sepsis mortality (Table 3). Consequently, HSP90α

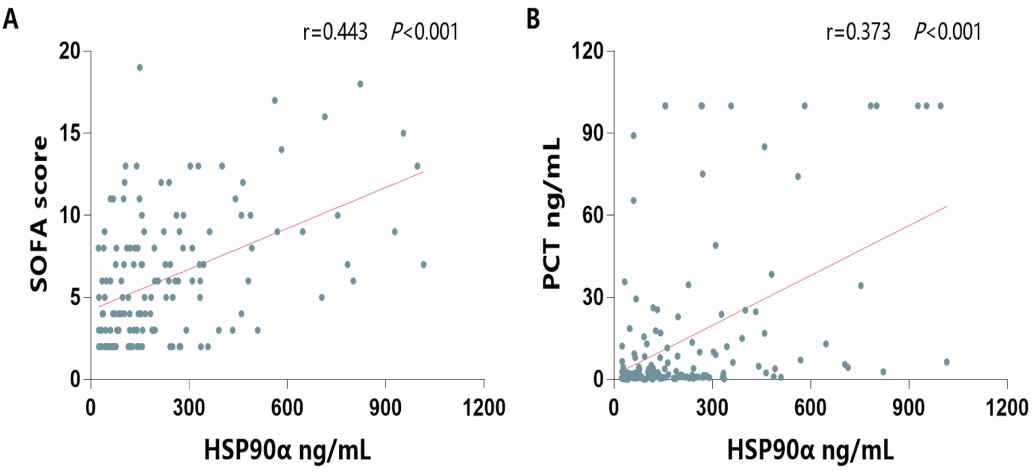

**Figure 3** Spearman correlation analysis of serum HSP90α levels with SOFA score and PCT.

| Risk Factors | Univariate Regression | |
| --- | --- | --- |
| | OR (95%CI) | *P*-Value |
| HSP90α | 1.302 (1.059,1.600) | 0.012 |
| SOFA | 1.236 (1.119,1.366) | <0.001 |
| PCT | 1.014 (1.002,1.226) | 0.024 |
| BUN | 1.046 (1.007,1.087) | 0.021 |
| Crea | 1.002 (1.000,1.004) | 0.034 |

**Figure 4** Univariate logistic regression analysis for sepsis-related factors.

**Table 3 Multivariate logistic regression analysis for sepsis after adjusting effects of confounders.**

| Multivariate logistic regression analysis | B | S. E. | Wald | *P*-value | Odds ratio | 95% CI |
| --- | --- | --- | --- | --- | --- | --- |
| Model 1 | 0.264 | 0.105 | 6.297 | 0.012 | 1.302 | [1.059–1.600] |
| Model 2 | 0.281 | 0.118 | 5.693 | 0.017 | 1.324 | [1.051–1.668] |
| Model 3 | 0.275 | 0.116 | 5.600 | 0.018 | 1.317 | [1.048–1.654] |
| Model 4 | 0.303 | 0.129 | 5.550 | 0.019 | 1.354 | [1.051–1.745] |
| Model 5 | 0.309 | 0.141 | 4.794 | 0.029 | 1.361 | [1.033–1.795] |

Note:
Model 1 included only HSP90α. Procalcitonin were added to Model 2. Model 3 was Model 2 with further adjustment for SOFA. Model 4 was Model 3 with further adjustment for creatinine. Model 5 was built on Model 4 and added urea nitrogen.

(unadjusted OR = 1.302, 95% CI [1.059–1.600], *P* = 0.012) was significantly linked to sepsis deaths. The addition of other cofounders, PCT (model 2, OR = 1.324, 95% CI [1.051–1.668], *P* = 0.017), SOFA score (model 3, OR = 1.317, 95% CI [1.048–1.654], *P* = 0.018), Crea (model 4, OR = 1.354, 95% CI [1.051–1.745], *P* = 0.019) and BUN (fully adjusted model 5, OR = 1.361, 95% CI [1.033–1.795], *P* = 0.029) only slightly enhanced this effect.

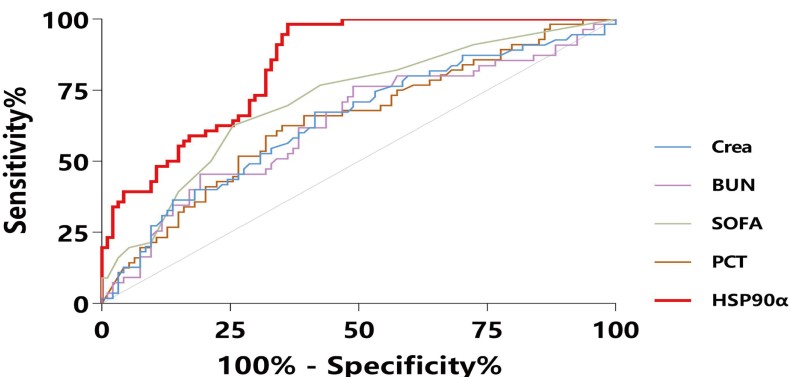

**Figure 5 ROC curves for diverse laboratory indexes on sepsis prognosis.** The *X*-axis refers to 1-specificity (false positive rate), and the *Y*-axis refers to sensitivity (true positive rate). Curves in different colors stand for diverse laboratory indexes. AUC means the area under curve, and when AUC approaches to 1 the prognostic performance on sepsis turns out to be excellent. Crea, creatinine; BUN, blood urea nitrogen.               

**Table 4 ROC analysis for laboratory indexes.**

|  | AUC (95% CI) | *z*-value | Youden index | *P*-value | Sensitivity (%) | Specificity (%) |
|---|---|---|---|---|---|---|
| HSP90α | 0.84 [0.77–0.90] | 11.00 | 0.62 | <0.001 | 98.2 | 63.8 |
| PCT | 0.64 [0.56–0.72] | 3.02 | 0.29 | 0.003 | 63.6 | 64.9 |
| SOFA | 0.71 [0.63–0.78] | 4.78 | 0.36 | <0.001 | 61.8 | 74.5 |
| Crea | 0.64 [0.55–0.71] | 2.80 | 0.25 | 0.005 | 67.3 | 57.6 |
| BUN | 0.63 [0.55–0.71] | 2.63 | 0.27 | 0.008 | 76.4 | 50.5 |

**Note:**
AUC, area under curve; CI, confidence interval; CRP, C-reactive protein; PCT, procalcitonin.

## The ability of HSP90α to predict the mortality

All sepsis patients were followed up for 42 days, and the mortality in 28 days was defined as the endpoint. To investigate the ability of HSP90α in predicting 28-day mortality in sepsis patients, the ROC curve (Fig. 5) was applied to calculated the cut-off value, sensitivity and specificity of PCT, SOFA score, Crea, BUN, and HSP90α as predictors of 28-day mortality (Table 4). For serum HSP90α level at admission, the AUC to predict 28-day mortality was 0.84 (95% CI [0.77–0.90], *P* < 0.001), with a 98.2% of sensitivity and a 63.8% specificity, and the optimal cut-off value is 120 ng/mL. The AUC of HSP90α is bigger than that of SOFA score (0.71, 95% CI [0.63–0.78], *P* < 0.001), PCT (0.64, 95% CI [0.56–0.72], *P* = 0.003), Crea (0.64, 95% CI [0.55–0.71], *P* = 0.005), and BUN (0.63, 95% CI [0.55–0.71], *P* = 0.008).

Moreover, a Kaplan–Meier survival analysis was used to estimate the effects of the HSP90α cut-off value (120 ng/mL) on survival rates in sepsis patients (Fig. 6B), with a comparison with higher SOFA score (≥5) (Fig. 6A). Of note, sepsis patients with serum HSP90α ≥120 ng/mL showed a significant increase in 28-day mortality (*P* < 0.001). However, higher SOFA score (≥5) had limited ability to predict 28-day mortality in sepsis

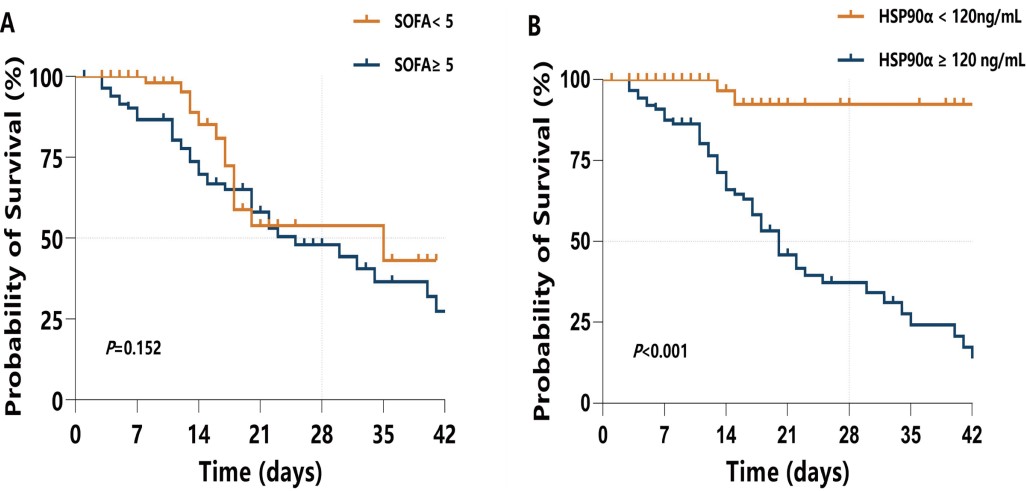

**Figure 6 Kaplan–Meier survival analysis presents the 28-d mortality of sepsis.** (A) When SOFA <5 or SOFA ≥5; and (B) when HSP90α <120 ng/mL or HSP90α ≥120 ng/mL.

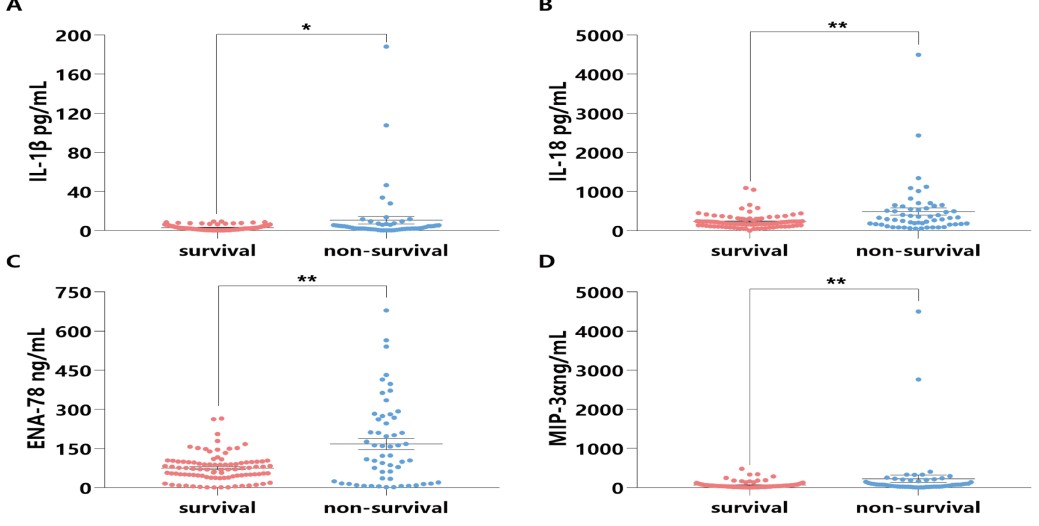

**Figure 7 Distribution and comparison for cytokines and chemokines in the Survival and Non-Survival groups of sepsis patients.** IL-1β (A), IL-18 (B), ENA-78 (C), MIP-3α (D); *P < 0.05, **P < 0.01.

($P = 0.152$). Collectively, these findings indicate that HSP90α has the valuable ability to predict the prognosis of the patients with sepsis.

## The potential mechanisms of HSP90α on survival in sepsis patients

To explore the potential mechanisms of HSP90α on survival in sepsis patients, some classical inflammatory cytokines (*i.e.* IL-1β, IL-18, ENA-78, and MIP-3α) associated with sepsis death were further tested. Notably, the expression levels of these four cytokines in the sera of the non-survival group were significantly higher than those in the survival group (all $P < 0.05$, Fig. 7).

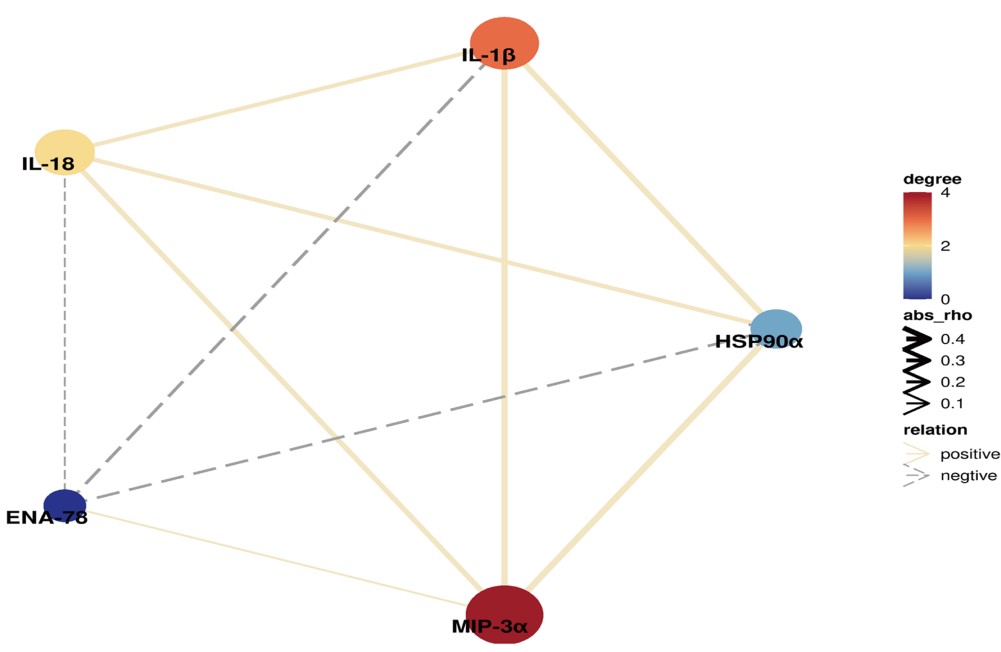

| | HSP90α | |
|---|---|---|
| | r | P-value |
| MIP-3α | 0.452 | <0.001 |
| IL-1β | 0.400 | <0.001 |
| IL-18 | 0.326 | <0.001 |
| ENA-78 | -0.155 | 0.059 |

**Figure 8 Correlation network of plasma HSP90α levels with laboratory indexes (Spearman analysis).**

Moreover, further spearman correlation analysis indicated highly significant correlations between HSP90α and IL-1β ($P < 0.001$, $r = 0.400$), IL-18 ($P = 0.01$, $r = 0.326$), and MIP-3α ($P < 0.001$, $r = 0.452$), except for ENA-78 ($P = 0.059$) (Fig. 8). These findings suggest that HSP90α may regulate the survival of sepsis patients *via* a vigorous cytokine expression.

## DISCUSSION

To date, no reliable biomarkers could provide sufficient power to predict early-onset and clinical outcomes in patients with sepsis. In present study, we have found that HSP90α may be a potential predictive biomarker for the diagnosis and prognosis of septic patients. Our studies have provided three lines of evidence supporting these findings: (i) serum HSP90α content was significantly higher in patients with sepsis than that of in healthy controls; (ii) compared with surviving septic patients, the elevated serum HSP90α levels were observed in the non-surviving ones; (iii) A higher level of serum HSP90α on admission could identify the 28-day mortality of patients with sepsis. In addition, we also found that provoking the secretion of pro-inflammatory cytokines and chemokines might be causes of HSP90α affecting the prognosis of sepsis patients.

Among the septic patients, the SOFA score serving as a major tool has been employed to indicate disease severity and predict mortality (*Gaini et al., 2019*; *Karakike et al., 2019*). However, the SOFA scoring is a system of fiddly calculation which involves multiple variables. Despite these issues, the SOFA score proved to be the best tool for the identification of patients with sepsis, which was in accordance with our results. Moreover, our study also indicated that serum HSP90α levels, with a high sensitivity of 88.67%, on ICU admission were superior in distinguishing sepsis compared to PCT (Fig. 1).

As mentioned previously, the SOFA score was a key predictive factor for ICU deaths (*Karakike et al., 2019*; *Raith et al., 2017*). In recent years, numerous biomarkers have been widely examined to improve the predicting ability of prognosis in patients with sepsis. The currently popular methods to identify sepsis clinical outcome are the utilize of several serum biomarkers, *i.e.*, CRP, PCT, SSA, lactate, and HBP, but the unsatisfied sensitivity and specificity limit their applications in clinical practice (*Pierrakos et al., 2020*). In present study, we have demonstrated that the 28-day mortality predictive ability of higher HSP90α contents ($\geq$120 ng/mL) was better than that of higher SOFA score ($\geq$5). The possible reasons for the discrepancy between our findings and other studies may due to different study subjects and follow-up time. Notably, circulating HSP90α possesses numerous strengths for its potential clinical use in the patients with sepsis. Firstly, serum HSP90α levels can be easily determined with high accurate by using ELISA technique, which can easily be popularized and applied in hospital laboratory. Secondly, due to the convenient detection of HSP90α, it is easy to achieve real time monitoring which aids to assess the septic patients' conditions dynamically and continuously. Finally, sepsis patients who died within 28 days had higher contents of serum HSP90α on day of ICU admission than did surviving patients, advocating HSP90α serving as a potential prognostic indicator.

As is well known, an inflammatory cytokine storm is a crucial pathogenic factor in patients with sepsis at the early stage. The elevated levels of IL-1β, IL-6, TNF-α, IL-18, and IL-37 in sepsis have previously been reported (*Ge, Huang & Yao, 2019*; *Lendak et al., 2018*; *Mierzchala-Pasierb et al., 2019*; *Song et al., 2019*; *Wu et al., 2021*). In other studies, the inflammatory cytokines blockade therapies appear to be beneficial for the sepsis treatment (*Saha et al., 2020*; *Xiong et al., 2020*; *Xu et al., 2018*). Given that HSP90α could regulate inflammatory response by engaging innate immune cells and activating downstream cytokines production, we determined serum levels of two classical inflammatory cytokines (IL-1β and IL-18), together with two chemokines (ENA-78, and MIP-3α) and identified that IL-1β, IL-18, and MIP-3α levels were positively related to those of HSP90α. Therefore, it was tempting to infer that HSP90α could be a potential molecule in the pathogenesis of sepsis.

## Limitations

Nevertheless, limitations of our study are noteworthy in the interpretation of results. First, this retrospective study might have caused selective bias resulting in overestimated the diagnostic and prognostic value of HSP90α. Second, the serum levels of HSP90α was not monitored at sequential time points after sepsis developed. Third, the follow-up period was short, and long-term survival was not analyzed. Fourth, the fact that present study

included a single center hindered our ability to draw conclusions. Fifth, a sample size estimation was not done. Quite evidently, additional studies are needed to verify these results.

## CONCLUSIONS

In current study, we provided evidence that serum HSP90α was a potential biomarker for evaluating the power of diagnosis and short-term prognosis of patients with sepsis, which could assist clinician identification of sepsis at early stage and lay the groundwork for the development of novel therapeutic target for patients with sepsis.

### Funding

This work was supported by the National Natural Science Foundation of China (Nos. 82160361 and 81960363), the Yunnan Fundamental Research Projects (202001BA070001-040 and 202001BA070001-055) and the Yunnan Health Training Project of High Level Talents (H-2019045). The funders had no role in study design, data collection and analysis, decision to publish, or preparation of the manuscript.

### Grant Disclosures

The following grant information was disclosed by the authors:
National Natural Science Foundation of China: 82160361 and 81960363.
Yunnan Fundamental Research Projects: 202001BA070001-040 and 202001BA070001-055.
Yunnan Health Training Project of High Level Talents: H-2019045.

### Competing Interests

The authors declare that they have no competing interests.

### Author Contributions

- Fuxing Li performed the experiments, analyzed the data, prepared figures and/or tables, and approved the final draft.
- Yulin Zhang performed the experiments, prepared figures and/or tables, and approved the final draft.
- Bocheng Yu performed the experiments, prepared figures and/or tables, and approved the final draft.
- Zihua Zhang performed the experiments, prepared figures and/or tables, and approved the final draft.
- Yujuan Fan analyzed the data, authored or reviewed drafts of the paper, and approved the final draft.
- Li Wang analyzed the data, prepared figures and/or tables, and approved the final draft.
- Mingjing Cheng analyzed the data, authored or reviewed drafts of the paper, and approved the final draft.

- Ping Yan conceived and designed the experiments, authored or reviewed drafts of the paper, and approved the final draft.
- Weidong Zhao conceived and designed the experiments, authored or reviewed drafts of the paper, and approved the final draft.

### Human Ethics

The following information was supplied relating to ethical approvals (*i.e.*, approving body and any reference numbers):

The First Affiliated Hospital of Dali University granted Ethical approval to carry out the study within its facilities (Ethical Application Ref: 20190612).

### Data Availability

The raw measurements are available in the Supplemental Files.

### Supplemental Information

Supplemental information for this article can be found online at http://dx.doi.org/10.7717/peerj.12997#supplemental-information.

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
