# Peer review of "Evaluation of the diagnostic and prognostic values of serum HSP90α in sepsis patients: a retrospective study"

_PeerJ, doi:10.7717/peerj.12997_

## Round 0.1 · original submission · Major Revisions

In your revised submission, please state clearly and identify where you provided the missing information identified by the reviewers and the new data that resulted from the analyses suggested by the reviewers.

·

Basic reporting

This manuscript of a good piece of work and adds significant information to the literature. The article structure is good and follows the journal`s pattern and guidelines. The hypothesis or study questions are well elaborated and tested. Thanks to the authors for sharing the raw data.

Experimental design

Thank you for providing the information on the ethical context of the manuscript. Since the authors adopted a case-control approach in this study, I did not find any information on sample size estimation? The representation of the sample for other settings is quite valuable, authors should discuss this in limitation if sample size estimation was not done.
It will be also valuable for readers to get details of the hospital in which this study is conducted i.e., type of hospital, the number of patients received or covered, no. of beds and types of services etc.
Please also include the normal values used in the current hospital for HSP90α or other biomarkers in the methodology section.

Validity of the findings

The authors have vigorously tested the relationship of HSP90α with sepsis and mortality. In the abstract section, the AUC of HSP90α was 0.8 while it has a different value in Figure 2.
Table 1 provides information on three arms of the study. Since the study design was case-control and the authors have matched all three arms at the time of patient selection, I did not see a statistical comparison of demographics between these three arms.
I have also observed that there are some missing data in the analysis, but the information of treating the missing values is not provided in the methodology section?
Table 1: 56/94???? what does it indicate, or the data was only available for 94 patients out of 150 septic patients? or this 94 are the patients who survived? the similar confusion will be created among readers so it is advised to please consider the elaboration of results presented in the table carefully. Technically, I will estimate the percentage of mortality i.e. 56/150 septic patients. How the data is presented in the table must be described in the footnotes of the table.
I am impressed by the logistic models in this study. However, I would suggest having values for univariate too for each included variable. If authors agree on that, they can provide univariate odds ratio and p values for each of the included variables in text and then can provide detailed information on the logistic regression.
Tabel 3: the information on Model 5 is not provided in the footnotes. I have also one concern here, the authors have aggressively analyzed the relationship between various markers with sepsis but the association of comorbidities is also questionable in this regard. Can authors discuss this in the discussion section to rule out any confounding impact of comorbid conditions?

·

Basic reporting

The manuscript entitled “Evaluation of the diagnostic and prognostic values of serum HSP90α in sepsis patients: a retrospective study” by Fuxing Li et al. investigated whether heat shock protein 90 α (HSP90α) can be used as diagnosis and prognosis markers in patients with sepsis. They have found that 1) serum HSP90α in sepsis patients were higher than those in ICU and healthy controls; 2) HSP90α was an independent risk factors for sepsis mortality; 3) HSP90α is a prognostic marker of sepsis.

Experimental design

In Figure 8, HSP90α is highly correlated to cytokines and chemokines (IL-1β, IL-18, and MIP-3α). If these factors along with HSP90α are assembled into a biomarker signature/model, will it demonstrate a better performance (increased AUC and OR) to diagnose and predict prognosis in septic patients? Authors can also test the performance of HSP90α combined with Crea, BUN, SOFA score and PCT.

Validity of the findings

1. In Figure 4, five variables were shown as significant risk factors for the survival of sepsis patients, including PCT, SOFA, creatinine (Crea), blood urea nitrogen (BUN) and HSP90α. Although ORs were statistically significant, how is it clinically important with an OR that is almost equals to 1? After adjusting multiple variables, OR was almost 1, as shown in Table 3.
2. To support the potential clinical application of HSP90α, the cut-off value (120 ng/mL) should be validated in independent patient cohorts
3. Authors indicated that HSP90α affected the prognosis of sepsis patients via provoking the secretion of pro-inflammatory cytokines and chemokines. However, the correlation between HSP90α level and inflammatory cytokines cannot support the conclusion that HSP90α induces the expression of cytokines thus leading to sepsis.

Additional comments

In text line 28, “blockage of HSP90α could attenuated the inflammatory damage”. It should be “attenuate”.

---

## Round 0.2 · accepted · Accept

Thank you for carefully addressing all of the reviewers' comments.

·

Basic reporting

All the comments are addressed by the authors.

Experimental design

All the comments are addressed by the authors.

Validity of the findings

All the comments are addressed by the authors.

Additional comments

All the comments are addressed by the authors.

·

Basic reporting

Authors have successfully responded to all my comments.

Experimental design

None.

Validity of the findings

None.

Additional comments

None.